# Training a Vision Transformer from scratch in less than 24 hours with 1 GPU

**Saghar Irandoust**
Borealis AI
saghar.irandoust@borealisai.com

**Thibaut Durand**
Borealis AI
thibaut.durand@borealisai.com

**Yunduz Rakhmangulova**
Borealis AI
yunduz.rakhmangulova@borealisai.com

**Wenjie Zi**
Borealis AI
wenjie.zi@borealisai.com

**Hossein Hajimirsadeghi**
Borealis AI
hossein.hajimirsadeghi@borealisai.com

## Abstract

Transformers have become central to recent advances in computer vision. However, training a vision Transformer (ViT) model from scratch can be resource intensive and time consuming. In this paper, we aim to explore approaches to reduce the training costs of ViT models. We introduce some algorithmic improvements to enable training a ViT model from scratch with limited hardware (1 GPU) and time (24 hours) resources. First, we propose an efficient approach to add locality to the ViT architecture. Second, we develop a new image size curriculum learning strategy, which allows to reduce the number of patches extracted from each image at the beginning of the training. Finally, we propose a new variant of the popular ImageNet1k benchmark by adding hardware and time constraints. We evaluate our contributions on this benchmark, and show they can significantly improve performances given the proposed training budget.

We will share the code in https://github.com/BorealisAI/efficient-vit-training.

## 1 Introduction

Recently, the Transformer architecture [1] has become a key ingredient of an impressive number of computer vision models [2, 3, 4, 5, 6, 7, 8, 9, 10, 11, 12]. However, training large Transformer models usually come at an enormous cost. For example, training a small vision Transformer (ViT) like DeiT-S takes about 3 days on 4 GPUs [4]. To reduce the cost, we propose to explore the following problem: *how to train from scratch a ViT model in less than 24 hours with a single GPU*. We think that progress in this direction could have a large impact on the future of computer vision research and applications for multiple reasons. **(1) Speed-up model development.** New models in ML are often evaluated in performance by running and analyzing experiments on them, which is not a scalable method when the training cost for each experiment is too high. By reducing the training cost we shorten the development loop. **(2) Be more accessible.** Most ViT models are trained from scratch by using multiple GPUs or TPUs, which unfortunately excludes researchers who do not have access to such resources from this area of research. By using only 1 GPU for our benchmark, we significantly reduce the training cost of ViTs which allows more researchers to push this research direction forward. **(3) Reduce the environmental cost.**

Has it Trained Yet? Workshop at the Conference on Neural Information Processing Systems (NeurIPS 2022).

One approach to reduce the training cost is to develop more efficient specialized hardware or more efficient data representations like half-precision. Another orthogonal approach is to develop more efficient algorithms. In this paper, we focus on the second approach. A lot of methods (*e.g.* pruning [13, 14, 15]) have been developed to reduce the inference cost, but a limited number of works are exploring ideas to reduce the training cost. [16, 17] explores how to train ViTs from scratch on small-size datasets. [18] explores how to train a BERT model on text data in 24 hours but it uses a server of 8 GPUs whereas we limit ourselves to a single GPU. Primer [19] proposes searching for more efficient alternatives to Transformer, but it focuses on NLP. We tried to apply the findings from this work to ViT, but we did not see any improvement. Hence, it remains unclear to us if the improvements developed for the NLP domain can also be generalized for computer vision applications.

We define our objective as gaining the highest performance metric within our fixed budget. To reduce the training cost, we propose two algorithmic contributions. First, we show that adding a locality mechanism in each feed-forward network of a Transformer encoder architecture can significantly improve the performance given a fixed resource budget. Second, we propose an image size-based curriculum learning strategy to reduce the training time per epoch at the beginning of the training. The training starts with small images, and then larger images are gradually added to the training. Beyond the algorithmic changes introduced to reduce the training cost, we also formally define our new benchmark on ImageNet1k by including resource constraints (1 GPU and 24 hour time budget), and evaluate our model on it.

## 2  Proposed method

### 2.1  Locality in vision Transformer architecture

In this section, we first explain the vision Transformer (ViT) architecture proposed in [3], and then we describe our changes to the architecture to speed-up the training.

**ViT architecture.**    The vanilla Transformer [1] receives as input a 1D sequence of token embeddings. To process 2D images, the ViT model [3, 4] splits each input image into a sequence of non-overlapping flattened 2D patches. The patches are mapped to $D$ dimensions with a trainable linear projection. The output of this projection is usually named patch embeddings. Then, the learnable position embeddings are added to the patch embeddings to encode positional information of each patch in the image. The output sequence of embedding vectors $z_0$ serves as the input to the Transformer encoder. The Transformer encoder [1] consists of alternating layers of multi-head self-attention (MSA) and Feed-Forward Networks (FFN). LayerNorm (LN) [20] is applied before every block, and residual connections after every block. For a Transformer encoder with $L$ blocks, the output representation is computed following the equations:

$$z_l' = z_l + MSA(LN(z_l)) \qquad z_{l+1} = z_l' + FFN(LN(z_l')) \qquad l \in \{1, \ldots, L\} \tag{1}$$

The FFN is composed of two linear layers separated by a GELU activation [21]. The first linear layer expands the dimension from $D$ to $4D$, and the second one reduces the dimension from $4D$ back to $D$.

**Locality in ViT architecture.**    The self-attention layer of a ViT captures global dependencies between all of the patches, but it lacks locality inductive bias, in particular a mechanism to allow information exchange within a local region. In order to introduce locality into vision Transformers, we only adapt the FFNs while the other parts, such as self-attention and position encoding, are not changed. We propose to add locality to the ViT architecture by adding a depth-wise convolution layer into each FFN. A $3 \times 3$ depth-wise convolution is added between the two fully-connected layers in the FFN (Figure 1). Before each $3 \times 3$ depth-wise convolution, a sequence-to-image (Seq2Im) layer is used to convert each flattened patch representation into a 2D patch representation. Similarly, an image-to-sequence (Im2Seq) layer is used to convert each 2D patch representation into a flattened patch representation. We also replace the GELU activation layer [21] by the h-swish [22].

**Connection with existing works.**    Other works [5, 23, 9, 10, 11] explore adding locality in ViT architecture. Most of them analyze the impact of locality mechanism on the final accuracy, but to the best of our knowledge, none of them study the impact of locality mechanism on the training speed. The closest work to our architecture is probably LocalViT [23], which also uses convolutions in the FFN. There are 3 main differences between LocalViT and our model. First, our architecture uses LayerNorm [20] as a normalization layer, whereas LocalViT uses 2D BatchNorm [24]. Second,

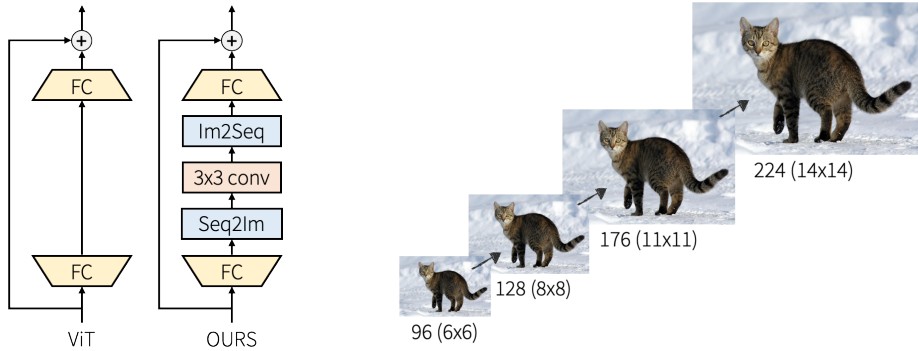

Figure 1: Summary of our two main contributions. **Left:** comparison between the feed-forward network in the vanilla ViT and our architecture. Our architecture adds locality by using a $3 \times 3$ depth-wise convolution added between the two fully-connected layers. Each patch is unflattened (resp. flattened) by the Seq2Im (resp. Im2Seq) layer. **Right:** the proposed image size-based curriculum learning. The model only sees small images (*e.g.* 96) at the beginning of the training, then the image size is gradually increased to reach the standard image size (*e.g.* 224).

the expanding and squeezing layers are implemented as fully-connected layer in our architecture, whereas LocalViT uses convolution layers. Finally, our architecture uses h-swish [22] as activation layer, whereas LocalViT uses a combination of h-swish and squeeze-and-excitation module [25]. We observe our contributions are important and lead to a more efficient architecture (see subsection 3.2).

## 2.2 Image size-based curriculum learning

Training vision Transformers is traditionally done by using mini-batches of $224 \times 224$ RGB images sampled uniformly from the training data. Each image is usually decomposed to non-overlapping $16 \times 16$ patches, so the input of vision Transformers is usually a sequence of 196 flattened patches. The complexity of a vanilla Transformer architecture [1] is quadratic to the sequence length (*i.e.* number of patches) due to the attention mechanism. In this section, we explore an approach to reduce the sequence length (*i.e.* number of patches) to speed-up training. We develop a small to large image size-based curriculum learning strategy [26], where shorter patch sequences are used at the beginning of the training.

The key idea of curriculum learning [26] is to start small and learn easier aspects of the task, then gradually increase the difficulty level. There are different ways to use curriculum learning, but a popular one is to start training with easy examples, and then gradually adding more difficult examples [27, 28]. We use the image size as a proxy of the image difficulty. At the beginning of the training, the vision Transformer model is trained with low resolution images, and then image resolution is gradually increased every few epochs. We do this by resizing the input images. Figure 1 shows different image sizes (*i.e.* steps of the curriculum learning) for a given image. In each epoch, all the images have the same size, but the image size can increase between epochs. Then, a critical question is how to design a good strategy to increase the image size. First, it is important to define the initial image size *i.e.* the image size for the first epoch. Then, it is important to control when the image size increases. We use a linear rule that increases the image size by $M$ pixels every $N$ epochs. In the experimental section, we analyze the impact of these hyper-parameters.

By construction, all the layers in the vision Transformer architecture, except the positional embeddings, can adapt automatically to multiple sequence lengths. The positional embeddings are updated by interpolation after each image size increase. To avoid to deal with partial patches, we only use image sizes that can be decomposed in $16 \times 16$ patches. Using multiple image sizes during training can also help to learn better scale invariant representations.

Table 1: Comparison of different models trained from scratch on ImageNet1k for a time budget of 24 hours with a single GPU. We also report the official result of DeiT-S when trained during 72 hours on 4 GPUs as an upper bound results.

|  | Top-1 Accuracy (%) | Budget |
|---|---|---|
| DeiT-S | 43.05 | 24h and 1 GPU |
| LocalViT-S | 27.89 | 24h and 1 GPU |
| Ours | **53.98** | 24h and 1 GPU |
| DeiT-S | 79.9 | 72h and 4 GPUs |

## 3  Experimental result

This section gives experimental results for image classification on ImageNet1k, which is a well-known benchmark widely used to evaluate image classification models. We show our results, and analyze the impact of our contributions. The implementation details can be found in Appendix A.

### 3.1  Results

We evaluate models trained from scratch and in less than 24 hours, with a single GPU on ImageNet1k. To the best of our knowledge, there are no existing results for this benchmark so we define some baselines. We trained DeiT-S and LocalViT-S models for 24 hours with a single GPU by using their official implementations. The results are summarized in Table 1. We observe our model outperforms other models by a large margin. There is an improvement of about 11 pt w.r.t DeiT-S, and about 26 pt w.r.t LocalViT. We also report the DeiT-S results by using the original resources (4 GPUs and 3 days) as an upper-bound. We can see there is still a large gap between this upper bound and our model, but we also see a large improvement w.r.t the baseline DeiT-S. These experiments show adding locality in the Transformer mechanism and using an image size curriculum learning help to speed-up training in a limited resource setting. The next sections analyze the impact of each of these contributions.

### 3.2  Analysis of the locality mechanism

In this section, we analyze solely the impact of our locality mechanism as proposed in subsection 2.1 (without using the image size-based curriculum learning) and compare with DeiT-S [4] and LocalViT-S [23]. Figure 2 shows the results on the validation set of ImageNet1k for each training epoch. Each model is trained for a maximum of 24 hours on a single GPU. Interestingly, we can see adding the locality mechanism does not significantly increase the training time per epoch w.r.t DeiT-S, but it increases the top-1 accuracy by 6.2 pt. We observe the training time per epoch of LocalViT is about

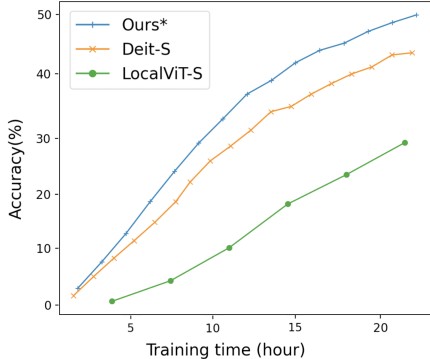
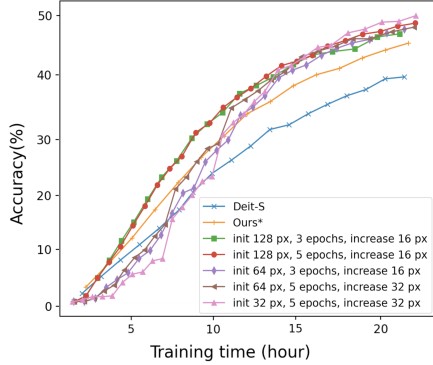

Figure 2: Top-1 accuracy on the validation set of ImageNet1k w.r.t the training time. Each mark indicates the performance at the end of each epoch. `Ours*` means our model without the image size-based curriculum learning. **Left:** Adding the locality mechanism increases performances but does not increase the training time per epoch significantly. **Right:** Using the image size-based curriculum learning improves the top-1 accuracy by about 4 pt, and it is quite robust to its configuration.

2.4 times our model's, so its training stops after 6 epochs. An advantage of our locality mechanism is that it can get good performance when using quite small batch sizes, because it uses LayerNorm instead of BatchNorm. We also found using h-swish instead of GELU improves the performance by about 1 pt. This analysis brings evidence that our locality mechanism can help speed-up training, and as a result reduce the training cost.

### 3.3 Image size-based curriculum learning

In this section, we analyze the image size-based curriculum learning introduced in subsection 2.2. The proposed curriculum learning has three important parameters: the initial image size, the image size increase, and the number of epochs between each image size increase. Figure 2 shows the results for different configurations. The best result was obtained by starting training from $32 \times 32$ pixel images, and increasing the image size along each axis by 32 pixels after each 5 epochs of training until we reach the final image size (224 pixels). We can see that adding the image size-based curriculum learning to our model on top of the locality mechanism improves the top-1 accuracy by about $4.5$ pt. Using small images allows training for more epochs because the training time per epoch is smaller.

## 4    Conclusion

Our experimental results on ImageNet1k bring evidence towards a positive impact of the locality mechanism and the image size-based curriculum learning to reduce the training cost of a vision Transformer model. It is likely that the findings can be generalized to other Transformer architectures but more experiments will have to be performed to validate this idea.

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

## A  Implementation Details

In this appendix, we describe the implementation details of our model training.

The ImageNet1k dataset [29] contains 1.28M training images and 50K validation images from one thousand classes. Otherwise specified, we report model performances on the official validation split to follow [23, 4] protocol. We use ImageNet1k because it is a well-known benchmark, which is widely used to evaluate image classification models. Our implementation is based on the `timm` library [30] and DeiT [4]. We use DeiT-S model as the main baseline in our experiments. To get fair comparison, we only compare Transformer with small architectures because they have similar complexity and number of parameters. Our training protocol is quite similar to DeiT, with some changes. The training continues until the 24 hours time limit is reached. The AdamW optimizer is used with a momentum of 0.9. The batch size is set to 64 and the learning rate is set to $1e^{-3}$. The model is trained with a cross-entropy loss and by using label smoothing. We do not perform learning rate warm-up. We use a Quadro RTX 5000 with 16GB, hosted in an internal cluster, to run our experiments. We resize the images by using a bilinear interpolation.

**Assets.**  In this project, we use the following existing assets:

- `timm` (`https://github.com/rwightman/pytorch-image-models`), Apache-2.0 license

- DeiT (`https://github.com/facebookresearch/deit`), Apache 2.0 license
- LocalViT (`https://github.com/ofsoundof/LocalViT`), MIT license
- ImageNet1k (`https://image-net.org/download.php`)

