# OpenReview forum: "Training a Vision Transformer from scratch in less than 24 hours with 1 GPU"
_NeurIPS.cc/2022/Workshop/HITY — HITY Workshop NeurIPS 2022_

### Official Review · Reviewer_phu1 · 2022-10-06
**Paper studying the best possible result achievable when training a Vision Transformer in 24 hours on a single GPU**

**Rating:** 1
**Confidence:** 3

**Review:**

The paper proposes a new benchmark: Training models from scratch on ImageNet, using only a single GPU and less than 24 hours of

The paper proposes a new benchmark: The target is to reach the highest top-1 accuracy on ImageNet1k after training for at most 24 hours using a single GPU.
They then describe two variations to current Vision Transformer training procedures. The resulting approach results in an improved performance on this benchmark.

The first modification is aimed at including locality in the ViT architecture by modifying the feed-forward part of the transformer with an additional convolutional layer.  The second modification introduces training with images of reduced resolution during the beginning of training.

The paper introduces an interesting problem of speeding up the training of Vision Transformers. The introduced modification show potential in the presented experiments.

Feedback:
- I wonder if it could be more beneficial to first introduce the new benchmark properly, before describing the "solution" to it.
- Do you have results on how well those speed advantages in the 24-hour regime translate to full training, e.g. whether there are also speed advantages when training to the accuracy of DeiT-S after 72 hours presented in Table 1?
- The presented method of "Image size-based curriculum learning" seems quite similar to "Progressive Image Resizing" (e.g. described here: https://docs.mosaicml.com/en/v0.9.0/method_cards/progressive_resizing.html#attribution). Could add a statement in the paper about how your method differs?
- In Figure 2 (right) you present image size-based curriculum learning with multiple hyperparameters. Which version was used for the results in Table 1?

Nits:
- You could compress sequential citations, e.g. in Line 15: "[2-12]" instead of listing them all.
- I am not sure that the statement in Line 95 that image size is a proxy for image difficulty is valid. Why should smaller images be easier? If anything, information is lost.
- Line 104: "except[s]"
- Line 110: "which is [a] well-known"
- Line 115: "there [are] no existing results"

---

### Official Review · Reviewer_TRAi · 2022-10-06
**Towards faster early-phase training of ViTs**

**Rating:** 1
**Confidence:** 3

**Review:**

The paper aims to improve the training of ViT models under a fixed computational budget (one GPU, 24 hours)---much smaller than what is usually used. Progress on this problem would enable shorter development cycles of new models, and make them more widely accessible to researchers with less compute hardware.

The work proposes modifications of the architecture (locality) and order in which data is presented to the model (curriculum learning) that speed up training. These modifications are evaluated, both jointly and independently, on a new benchmark derived from ImageNet1k by addition of constraints on the computational budget. Results indicate that both mechanisms help achieve faster training within the given budget.

(I am not very familiar with the architectures used in the paper, hence the low confidence score.)

---

Detailed comments:

Some of the architecture modifications are motivated, but exchanging the GeLU activations seems arbitrary. It would be good if the authors could provide a motivation in section 2 how this modification helps to speed up training.

In section 3.1 it would be helpful to add a remark about how the training hyperparameters were chosen.

A satisfying extension of Figure 2 (left) would be to extend these plots to times longer than 24 hours. This should be doable with little effort. If the proposed model reaches the final performance of DeiT-S in less time, this would further strengthen the paper's contribution.

I also think that the manuscript could benefit from a short "Conclusion" section at the end where the most important findings are restated. This will make the paper easier to skim.

---

Miscellaneous comments:

- L104: "excepts" → "except"
- L110: Missing article before "well-known"
- L115: "is" → "are"

---

### Official Review · Reviewer_uh8M · 2022-10-17
**Faster ViT training**

**Rating:** 1
**Confidence:** 3

**Review:**

The paper suggests improving the speed of ViT training by adding convolutional layers and resolution based curriculum learning.  A benchmark for resource limited ViT training is also proposed.  The speed-ups seem to beat baselines in the 24 hour 1 GPU timeframe proposed in the benchmark, but the performance in this setting is very far from what is achievable with 4 GPUs in 72 hrs, so the paper seems to oversell significantly.

---

### Decision · Program_Chairs · 2022-10-20

Accept